# Microcrack evolution and permeability enhancement due to thermal shocks in coal

Hongtu Zhang[1,2], Dengke Wang[1,2]*, Chong Yu[1,2], Jianping Wei[1,2], Shumin Liu[3], Jianhua Fu[1,2]

1 State Key Laboratory Cultivation Base for Gas Geology and Gas Control, Henan Polytechnic University, Jiaozuo, China, 2 School of Safety Science and Engineering, Henan Polytechnic University, Jiaozuo, Henan, China, 3 State Key Laboratory of Coal Mine Disaster Dynamics and Control, College of Resource and Safety Engineering, Chongqing University, Chongqing, China

* wdk@hpu.edu.cn

**Data Availability Statement:** All relevant data are within the paper and its Supporting Information files.

**Funding:** This work was funded by the National Natural Science Foundation of China (No.

## Abstract

To understand the effects of thermal shock on microcrack propagation and permeability in coal, thermal shock tests were conducted on coal specimens by using a constant temperature drying oven (105 ˚C) and a SLX program controlled cryogenic tank. The growth and propagation of microcracks were measured with computer tomography (CT) scanning and scanning electron microscope (SEM) tests. Results showed that thermal shocks improved the permeability of coal significantly. Notably, the permeability of coal after thermal shocks increased from 211.31% to 368.99% and was positively correlated with temperature difference. CT scanning images revealed that thermal shocks increased the crack number, crack volume and crack width as well as smoothened and widened the gas flow paths, thereby enhancing coal permeability. Moreover, SEM images showed that heating-cooling shocks created more new microcracks, forming more complex crack propagation paths and better connectivity among microcracks in coal compared to cooling shocks. We proposed a crack propagation criterion for coal to explain the mechanism of crack failure and propagation during thermal shocks. Our experiment results and theoretical analysis indicate that the heating-cooling shock is more effective in damaging and breaking coal than the cooling shock. Thus, it can be used as an alternative approach to enhance coal permeability in the production of coalbed methane (CBM).

## 1. Introduction

Coalbed methane (CBM) is an important resource of energy in China. It is viewed an environmentally friendly source of energy. The key to improve the production of CBM is increasing coal permeability [1, 2]. Many macro-, meso-, and micro-pores coexist in coal, where micro-pores are the main points of methane adsorption. Nevertheless, CBM is very difficult to be drained without external disturbances because methane molecules can tightly adsorb on the micro-pore surface. Over the past decade, many studies have focused on waterless thermal-

51774118, 51904095), the Open Fund of State Key Laboratory of GeoMechanics and Deep Underground Engineering, China University of Mining & Technology (SKLGDUEK1814), the Program for Leading Talents in Scientific and Technological Innovation of Henan Province (204200510032), the Scientific and technological projects of Henan Province (192102310200) and the Natural Science Foundation of Henan Polytechnic University (J2018-1).

**Competing interests:** The authors have declared that no competing interests exist.

shock fracturing techniques [3–5], because hydraulic fracturing technology consumes a large volumes of water and may affect public health and the environment [6].

As a prospective CBM exploitation method, thermal shock fracturing technique has received increasing attentions. Liquid nitrogen ($LN_2$) freezing is reported to be an effective fracturing method due to its extremely low temperature, about -196 ˚C at atmospheric pressure. Most recent studies have primarily focused on the physico-mechanical properties, permeability features, fractures initiation and growth induced by $LN_2$ cooling shocks. Li et al. [7] proposed a new $LN_2$ gasification fracturing technology which involved $LN_2$ fracturing, rock contraction fracturing, rock embrittlement fracturing and nitrogen expansion fracturing. Cai et al. [4, 8, 9] studied the change in pore structure, permeability, macro-mechanical properties and cracking features in coal and shale specimens after $LN_2$ cooling. The results showed that $LN_2$ cooling augmented the propagation of micro-cracks in specimens, and resulted in the deterioration of mechanical properties and the improvement of permeability of specimens. For example, the coal permeability increased by 48.89%–93.55% after $LN_2$ cooling treatment, and the compressive strength of cool-treated coal specimens decreased by 16.18%–33.74% compared with intact specimens [4]. Under laboratory conditions, Qin et al. [5, 10] studied the effect of $LN_2$ freezing on pores and fracture structures in coal. After $LN_2$ freezing-thaw cycles, they found that the coal specimens' elastic moduli, uniaxial compressive strengths (UCSs) and longitudinal wave velocities dropped while the porosities and Poisson's ratios increased. Yin et al. [11] studied the permeability evolution and mesoscopic cracking behavior of coal under $LN_2$ cooling treatments, and found that the coal permeability and fracture expansion increased with increase in water content. Furthermore, Zhao et al. [12] studied the damage effect of $LN_2$ cooling on fracture propagation in coal by using wet and dry coal specimens. Their result showed that the porosity of wet coal specimens increased with the increase of water saturation, because $LN_2$ cooling led to the expansion and connection of fissures in coal. Zhang et al. [13, 14] analyzed fracturing capacities and thermal-cracking behaviors induced by $LN_2$ jet. They found that the heating process has a slight effect on generating thermal-crack compared to the $LN_2$ impact. Moreover, the fracturing capacity of $LN_2$ jet was better than those of abrasive water jet and supercritical $CO_2$ jet.

Heat injections can also enhance CBM recovery [15–17]. For instance, Huang et al. [18] reported that coal permeability increased by 2.2 times after 300 seconds microwave radiation at 2.45 GHz and 500 W. Based on the change of physico-mechanical properties due to heating treatment, many studies have attempted to explain the mechanism of improving CBM production. At adsorption equilibrium states, it is clear that the adsorption capacity of methane is sensitive to temperature variation [19]. Under the influence of heating treatment, the capacity of coals to adsorb methane decreased with increasing temperature [20–22], and the adsorption time increased with temperature under the same gas and pressure condition [22]. The temperature influence on gas flow always becomes very complicated due to the anisotropy and multi-mechanism of gas flow in coal [23, 24]. The relationship between coal permeability and effective stress, pore pressure and temperature was experimentally investigated [25–29]. It was found that the change of coal permeability with temperature variation primarily depended on the interaction of thermal and effective stresses, and was related to coal deformation. In addition, some permeability formulas combined with temperature and effective stress were successfully proposed to predict coal permeability under different conditions [30–32]. Heat injections increased coal permeability because in coal, sharp temperature change possibly leads to uneven thermal deformation and microcrack generation [33–36]. The expansion and connection of microcracks induced by heat injection boosted gas flow room and smoothened gas flow paths, leading to the increase of coal permeability [36–38]. Besides, the propagation of microcracks caused coal damages and degeneration of coal mechanical properties [39–41].

Scanning electron microscope (SEM) and computer tomography (CT) scanning technologies are effective means to investigate the growth and expansion of cracks in geological materials under thermal treatments [42, 43]. Cai et al. and Qin et al. studied the crack propagation in coal using SEM tests and found that $LN_2$ freezing stretched and increased the number of cracks, and accordingly improved the initial damage degree of the coal [5, 8–10]. The change in coal pore structure under increasing temperatures was investigated using SEM technology [44–47]. The results showed that, after heat treatments, the pore structure was significantly altered and additional micro-porosity developed. Yu et al. [48] investigated the micro-pore development in coal under high temperature using CT scanning experiments, and concluded that the change of micro-pores varies at different temperature ranges. Using CT scanning technique, Fan et al. [49] declared that the micro-porosity of granite increases with the increase in temperature from 400 ˚C to 800 ˚C, and the granite heterogeneity and anisotropy were mainly dominated by initial cracks below 200 ˚C. Based on CT scanning images, Kumari et al. [50] concluded that the injection of cold water causes thermally-induced inter- and intra-crystalline cracks in hot rocks, and then results in a porous zone with a porosity range of 2% to 3%.

From the above, there are few studies on permeability enhancement and thermal damage in coal and rock induced by heating—cooling treatment. Wu et al. [51, 52] used shale and granite specimens to study the variation of physico-mechanical properties under heating-cooling treatment. Their results demonstrated that the permeability of heated specimens significantly increased after $LN_2$- cooling treatments, while the density, P-wave velocity, strength and elastic modulus decreased. Wang et al. [53] confirmed that the coal permeability increased by 469.24% under heating-cooling shocks of 100 ˚C to -196 ˚C. Besides, it is necessary to understand the efficiency and mechanism of permeability enhancement by thermal shocks because larger coal permeability leads to higher CBM production. In this present study, the microcrack characteristic and permeability variation under thermal shocks were investigated. The main objectives of this study were (i) to determine the change characteristic of coal permeability under heating-cooling shocks using permeability tests, (ii) to determine the evolution of microcracks and damages in coal under thermal shocks through CT scanning and SEM tests, and (iii) to understand the mechanism of damage occurrence and permeability improvement in coal by proposing a propagation criterion of microcracks.

## 2. Experiment set-up

### 2.1. Coal specimens preparation

All testing specimens were made of fresh raw coal blocks which were collected by ourselves from Jiaozuo mining area of Henan Province in China. The sampling site is located at 35.346 N and 113.471 E. There are no conflicts of interest between us and the mining area, thus, no specific permissions were required for sampling. The coal blocks were processed into $\phi 50$ mm × 50 mm coal specimens for permeability tests and CT scans. The prepared coal specimens were placed in a drying oven at temperature of 60 ˚C for dry-seal preservation. During tests, the coal specimens were divided into four groups: A, B, C and D, with three coal specimens in each group. The physical parameters and the proximate analysis results of coal specimens are listed in Tables 1 and 2, respectively.

Similarly, granular coal specimens used for SEM tests were divided into four groups: E, F, G and H, with 3 mm to 6 mm in diameter sizes. Coal particles in groups E and G belong to anthracite coal, and coal particles in groups F and H belong to bituminous coal. The coal particles in groups E and F were subjected to cooling shocks, while the coal particles in groups G and H were subjected to heating-cooling shocks.

**Table 1. Specimen parameters and numbers.**

| Group No. | Specimen | Volume (cm$^3$) | Mass (g) | Apparent density (g/cm$^3$) | Temperature range |
|---|---|---|---|---|---|
| A | A1 | 98.196 | 142.513 | 1.451 | From 90 ˚C to -90 ˚C |
|   | A2 | 97.055 | 141.434 | 1.457 |  |
|   | A3 | 95.663 | 141.650 | 1.481 |  |
| B | B1 | 97.000 | 144.341 | 1.488 | From 120 ˚C to -'120 ˚C |
|   | B2 | 96.417 | 142.275 | 1.476 |  |
|   | B3 | 95.662 | 139.863 | 1.462 |  |
| C | C1 | 95.836 | 140.741 | 1.468 | From 150 ˚C to -150 ˚C |
|   | C2 | 98.428 | 142.442 | 1.447 |  |
|   | C3 | 97.616 | 142.489 | 1.459 |  |
| D | D1 | 95.296 | 141.708 | 1.487 | From 180 ˚C to -180 ˚C |
|   | D2 | 97.616 | 144.490 | 1.480 |  |
|   | D3 | 98.002 | 142.346 | 1.452 |  |

To avoid frost heaving effect from internal moisture during thermal impacts, all examined coal specimens were thoroughly dried using an oven at 105 ˚C for 24 h. The sampling locations are shown in Fig 1, and the proximate analysis results of coal particles are listed in Table 2.

## 2.2. Experiment process

In this study, as shown in Fig 2, a phoenix v | tome | x s industrial CT scanning system from General Electric Company (GE) in United States was used to scan the coal specimens. This industrial CT scanning system is equipped with two X-ray sources to meet the scanning accuracy of the different size specimens. The main technical parameters are listed in Table 3. To obtain high-quality and clear CT scan images, the specific parameters of scanning voltage, scanning electric current, exposure time, projection number and scanning time were set at 150 kV, 110 μA, 1000 ms, 1500 and 34 minutes, respectively. Besides, a triaxial gas seepage equipment was used to measure the permeability of the coal specimens. A coal specimen holder, an axial pressure loading pump, a confining pressure loading pump and a gas pressure loading device are the main parts of the triaxial gas seepage equipment. The main parameters are axial stress, confining pressure, gas pressure and gas flux. The maximal axial stress is 50 MPa with a full-scale accuracy of less than 0.5%, the maximal confining pressure is 20 MPa with a full-scale accuracy of less than 0.5%, the maximal gas pressure is 10 MPa with a full-scale accuracy of less than 0.2%, and the maximal gas flux is 500 ml/min at standard atmospheric pressure. The detailed procedures of permeability tests and CT scans under thermal shocks are as follows:

1. A prepared coal specimen was fitted into the coal specimen holder, and two metal loading rods were tightly attached to the two ends of the coal specimen, respectively. An axial

**Table 2. Proximate analysis and density.**

| Group No. | Coal rank | Proximate analysis | | | True density |
|---|---|---|---|---|---|
|   |   | $M_{ad}$ (%) | $A_{ad}$ (%) | $V_{daf}$ (%) | (g/cm$^3$) |
| A, B, C, D, E and G | anthracite | 0.92 | 16.7 | 6.43 | 1.59 |
| F and H | bituminous coal | 1.10 | 16.62 | 19.00 | 1.45 |

$M_{ad}$ is moisture; $A_{ad}$ denotes ash yield; $V_{daf}$ means volatile matter.

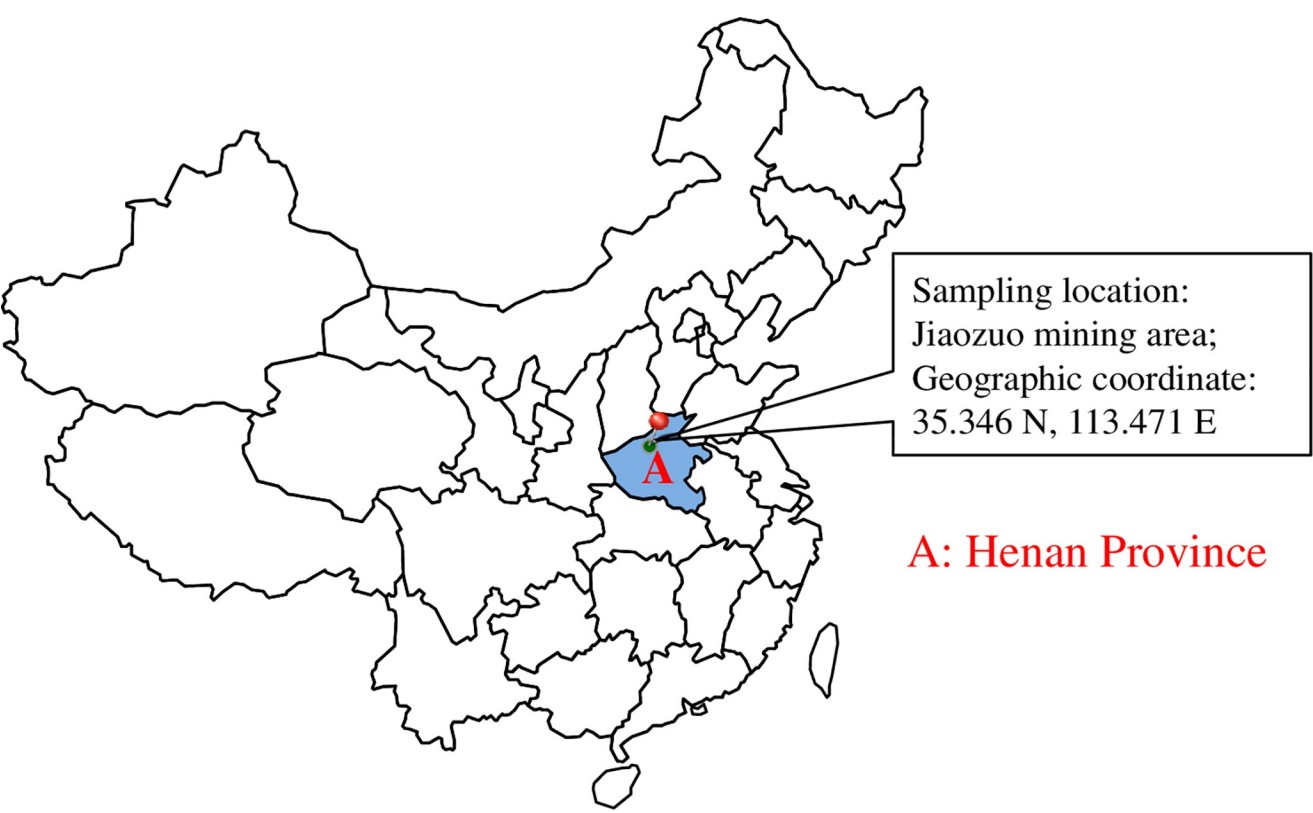

**Fig 1. The sampling locations of coal specimens in China.**

pressure of 1 MPa and a confining pressure of 1 MPa were loaded onto the coal specimen by using two high pressure pumps.

2. The main experiment parts were interconnected by some stainless steel tubs, and the gas outlet was closed. Helium gas at 0.5 MPa pressure was injected into the experiment system to check the air tightness. If the helium gas pressure did not drop after 6 hours, the experiment system was airtight and could be used to carry out the gas seepage tests.

3. The helium gas in the experiment system was released after the gas inlet was closed and the gas outlet was opened. The experiment system was vacuumed for 12 hours using a vacuum pump to remove the air and the helium gas.

4. After removing the vacuum pump, a high methane tank was connected with the gas inlet. Methane gas was injected into the coal specimen by opening the switch of the high methane tank. The pressure of the injected methane gas was regulated at 0.3 MPa via a regulator. The methane gas permeated through the coal specimen after the gas outlet was opened, and a mass flow meter recorded the methane gas flux. The permeability of the coal specimen was calculated using the methane gas stable flux.

5. When the gas seepage test was ended, the coal specimen was removed from the triaxial gas seepage equipment and then placed in the industrial CT scanning system for CT scanning test. After a CT scanning test, CT scanning images were obtained and were used to show the initial microcrak structures in the coal specimen.

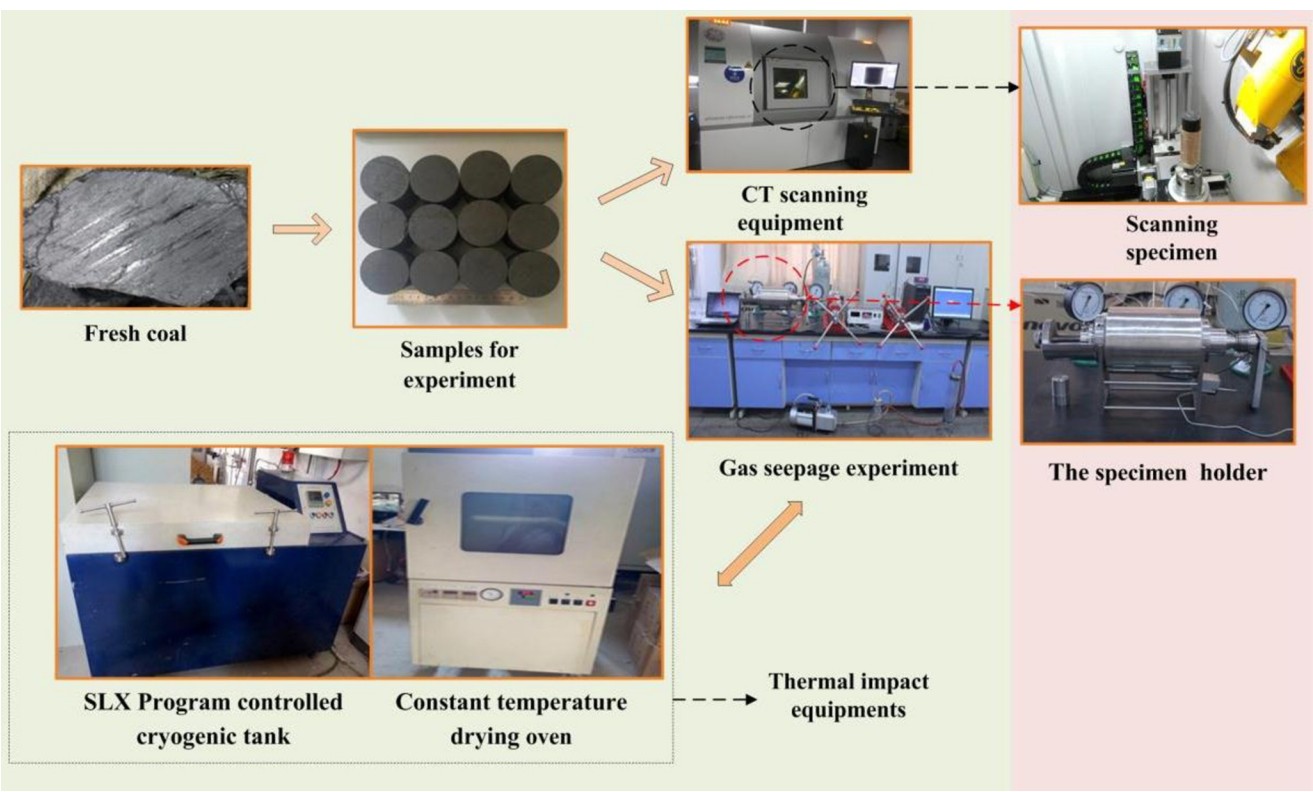

**Fig 2. The test process of permeability changes and internal crack evolution under thermal shocks.**

6. The coal specimen was subjected to a thermal shock after the first CT scan. After the thermal shock, the coal specimen was rescanned to obtain the CT scanning images. The comparison of CT scanning images before and after the thermal shock revealed change in microcrack structures in the coal specimen due to thermal shock.

7. After the thermal shock, the permeability of the coal specimen was remeasured by repeating the above steps (1)—(4) to obtain the permeability change from thermal shock.

Herein, taking the thermal shock test at 90 ˚C to -90 ˚C temperature range as an example to describe the experiment steps as follows: The volume and weight of each coal specimen were first measured and recorded three times. A constant temperature drying oven opened and the

**Table 3. The main technical parameters of X-ray sources.**

| Parameters | Micro-focus X-ray source | Nano-focus X-ray source |
|---|---|---|
| Voltage | From 10 kV to 240 kV | From 10 kV to 180 kV |
| The minimum size of focus | Less than 3 μm | Less than 1 μm |
| The maximum power | Larger than 320 W | Larger than 15 W |
| The minimum distance between the focus and the specimen | Less than 4.5 mm | Less than 0.35 mm |
| Resolution | Less than 2 μm | Less than 0.5 μm |
| Electric current | From 0.01 Am to 3.0 Am | From 0.01 Am to 0.88 Am |

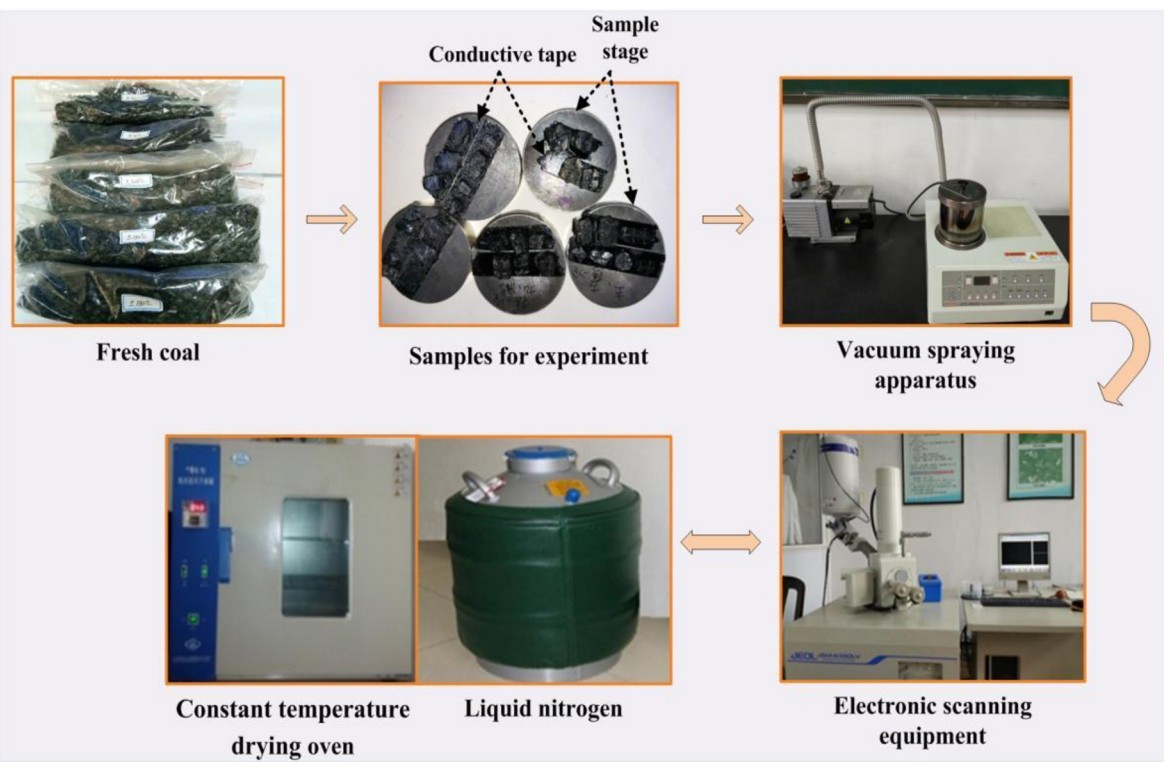

**Fig 3. The SEM test process of coal particles under thermal shocks.**

internal chamber temperature kept at 90 ˚C. Coal specimens were placed into the constant temperature drying oven and subjected to a heating shock for 2 h. After the heating shock, the coal specimens were quickly moved to the internal chamber of a SLX program controlled cryogenic tank at -90 ˚C for a 2 h cooling shock. Finally, after a full heating-cooling shock treatment, the volume and weight of each coal specimen were re-measured three times.

As shown in Fig 3, a JSM-6390LV SEM apparatus was used to check the development and distribution of surface fractures of coal particles before and after the temperature shocks. The amplification of this equipment is from 30 to 300,000 times and the best scanning precision is 3 nm under vacuum conditions. SEM test procedures are shown in Fig 3 of the coal particles under thermal shocks are as follows:

1. Before carrying out thermal shocks, the coal particles selected from groups E, F, G and H were fixed on testing platforms using conductive tape.

2. The testing platforms with coal particles were placed in the gold-sputtering zone of the vacuum spraying device for 30 minutes' surfaces gold sputtering. After gold-sputtering, the coal particles were placed on the objective table of the JSM-6390LV SEM equipment to locate and measure initial surficial microcracks.

3. Coal particles in groups E and F were suddenly cooling-treated with LN2 (-196 ˚C) for 2 h. Coal particles in groups G and H were subjected to a heating shock at 100 ˚C for 1 h and then an immediately cooling shock with LN2 for 1 h.

4. After the temperature shocks, the coal particles were rescanned by repeating step (2) to obtain the features of surficial microcracks of the coal particles.

5. The change characteristic of coal particles surficial microcracks in groups E, F, G and H were obtained by comparing the corresponding SEM images before and after thermal shocks.

## 3. Results and discussion

### 3.1. Coal permeability properties under thermal shocks

Wang et al. [54] comprehensively described seepage experiment process. The dynamic viscosity of test nitrogen was 17.812 μPa·s. The axial stress and confining pressure were set at 1 MPa. The inlet gas pressure $P_i$ was 0.3 MPa, and the outlet gas pressure $P_e$ was 0.1 MPa. For steady gas flow, the basic formula for calculating coal permeability is expressed as follows [55]:

$$K_g = \frac{2Q_0 p_0 L \mu}{A(p_i^2 - p_e^2)} \tag{1}$$

where $K_g$ is the gas permeability; $Q_0$ denotes the gas volume flow under standard conditions; $\mu$ represents for the experiment gas dynamic viscosity; $L$ represents the porous medium length; $A$ is the porous medium cross-sectional area; and $P_0$, $P_i$, and $P_e$ are the standard conditions of the gas pressure, inlet pressure and outlet pressure, respectively.

Coal specimens in group A were subjected to the thermal shock of 90 ˚C to -90 ˚C, at a temperature difference of 180 ˚C; coal specimens in group B experienced the thermal shock of 120 ˚C to -120 ˚C, at a temperature difference of 240 ˚C; coal specimens in group C underwent the thermal shock of 150 ˚C to -150 ˚C, at a temperature difference of 300 ˚C; and coal specimens in group D were subjected to the thermal shock of 180 ˚C to -180 ˚C, at a temperature difference of 360 ˚C. The permeability of coal specimens was calculated before and after thermal shocks using (Eq 1) under the 90 ˚C to -90 ˚C, 120 ˚C to -120 ˚C, 150 ˚C to -150 ˚C and 180 ˚C to -180 ˚C temperature conditions. In addition, the increase of each group's average permeability was calculated (see Fig 4).

As shown in Fig 4, the average increase of coal permeability in groups A, B, C and D were 211.31%, 258.01%, 314.01% and 368.99%, respectively. This result indicates that the average increase of coal permeability increased with temperature difference. Therefore, it can be inferred that thermal shocks caused expansion and contraction of coal matrix, subsequently inducing major changes in the internal structure and mechanical strength of coal specimens [56]. In addition, with the increase of temperature difference, a higher thermal stress caused a bigger internal structure damage, leading to a more serious degradation of mechanical strength and a larger increase in permeability of the coal specimens.

### 3.2. CT scanning results under thermal shocks

CT scanning is quite a fast, comprehensive and nondestructive technology [42, 43]. The scanning resolution of the industrial CT scanning system depends on the size of coal specimens. In this paper, the scanning resolution of CT apparatus is about 25 μm for the pillar coal specimens. From the result of seepage experiments results, thermal shocks increased the coal permeability, but the actual evolution process of microcracks in coal was not visible. This can be realized by using the industrial CT scanning system. Through CT scanning, digital models of the coal specimens can be reconstructed using 3D reconstruction function of the industrial CT scanning system. In addition, the detailed information of pores, throats and cracks in the coal specimens can be analysed by the feature extraction function. Coal specimens A1, B1, C1 and D1 were selected to be successively scanned before and after thermal shocks. The scanning

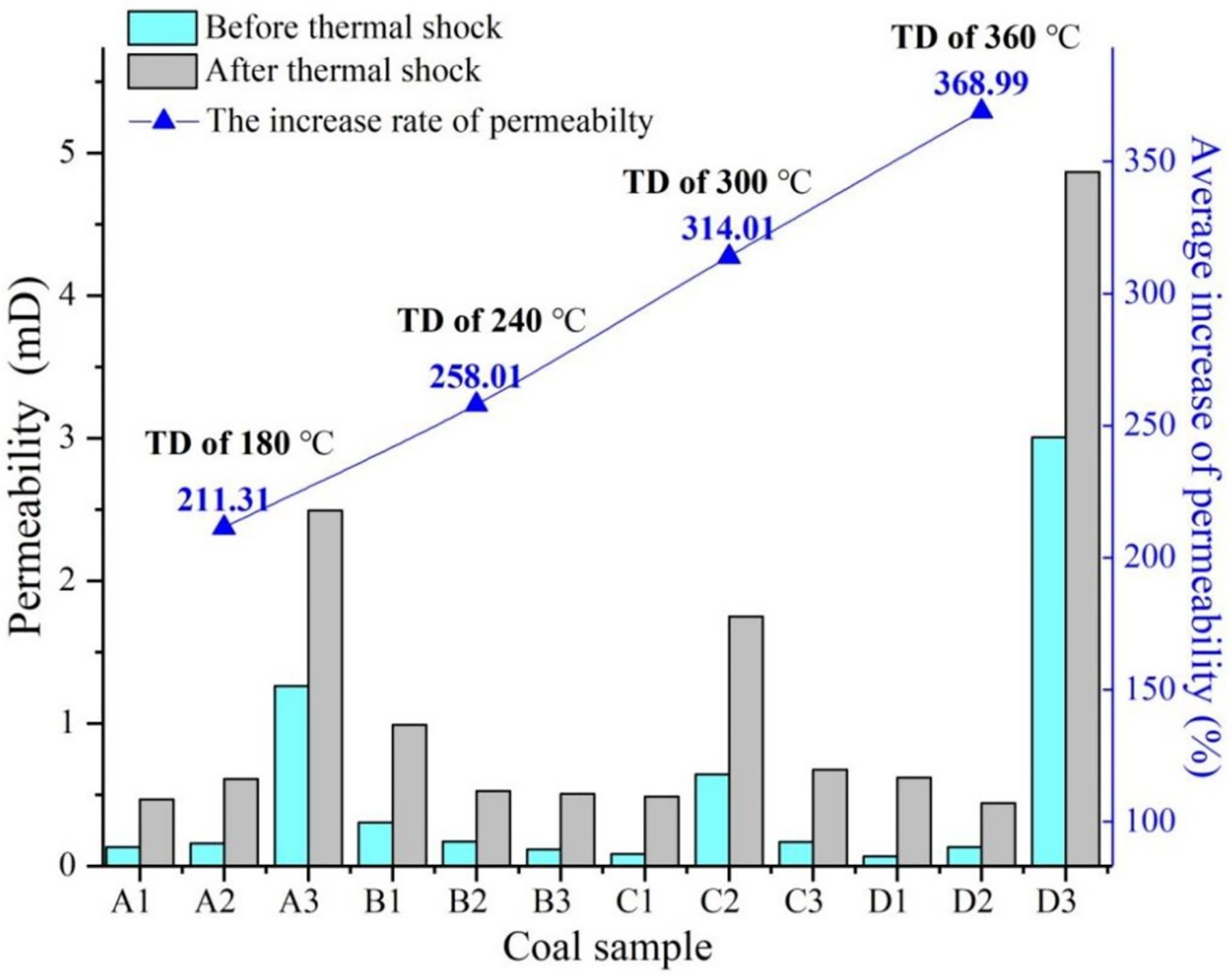

**Fig 4. Effect of thermal shocks on coal permeability (TD of 180 ˚C means the temperature difference of 180 ˚C).**

data was processed and analysed using the VG Studio MAX software. Fig 5 shows the reconstructed 3D crack structures in the coal specimens before and after the thermal shocks.

In specimen A1, the volume of microcracks increased from 13.730 mm$^3$ to 23.628 mm$^3$, a 72.09% increment. After thermal shock, the maximal width of microcracks was 70 μm, 75% increment than before thermal shock. Correspondingly, both the volume and the maximal width of microcracks increased. Before thermal shocks, original cracks were flaky, slightly curled, characterized by a slope of approximately 80˚. Under thermal stresses, a new microcrack was generated close to the bottom of specimen A1 and connected to the original crack (see Fig 5a).

The development of initial microcracks in specimen B1 is stronger than that in specimen A1. The crack network of specimen B1 was made up of a large inclined main crack and a series of small cracks. Before thermal shocks, the initial crack volume was 20.917 mm$^3$ and the maximal crack width was 40 μm. After heating-cooling shock, the cracks further expanded and interconnected, finally forming a stereoscopic A-shaped main crack network (shown in

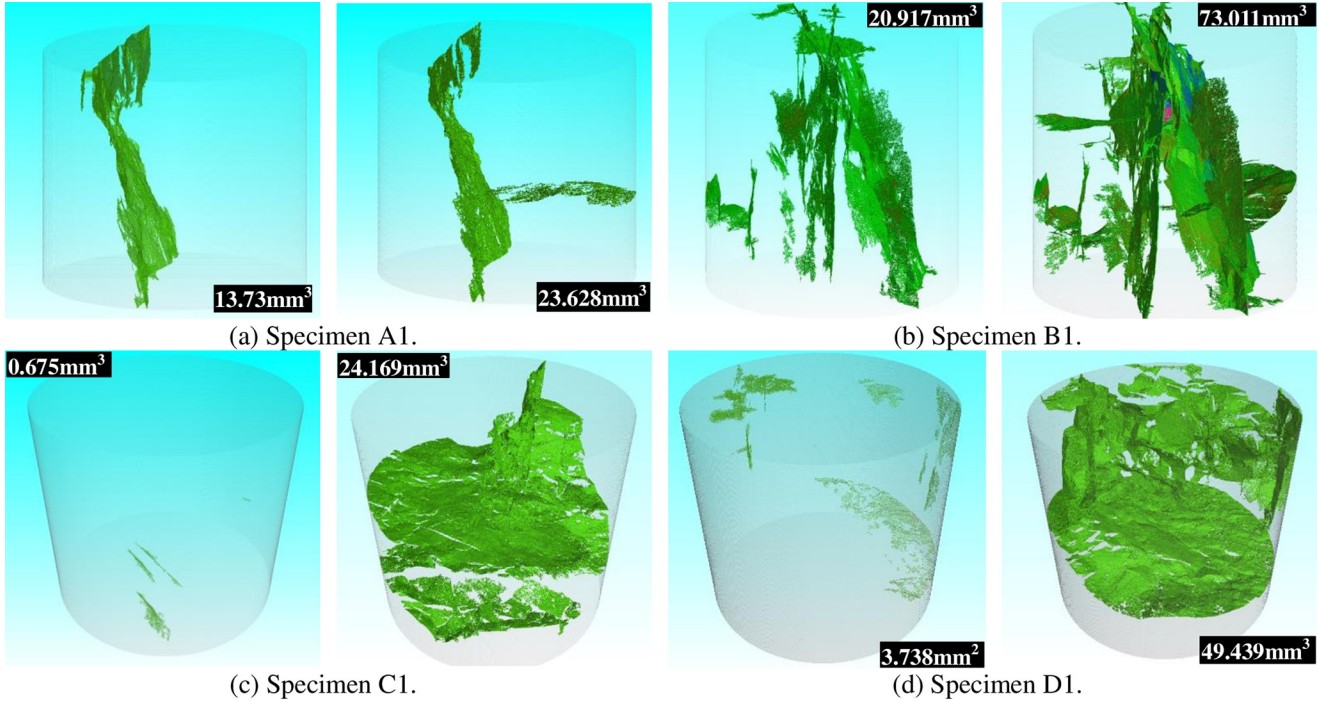

**Fig 5. The reconstructed 3D crack structures in the examined coal specimens before and after thermal shocks.**

Fig 5b). The crack volume and the maximal crack width increased to 73.011 mm$^3$ and 90 μm in increments of 249.05% and 125.00%, respectively.

Before heating-cooling shock, the 3D rebuilt digital model of specimen C1 indicates that there were three little scattered microcracks at the bottom of specimen C1. The initial crack volume and the maximal crack width were 0.675 mm$^3$ and 40 μm, respectively. After heating-cooling shock, a cap-shaped crack was formed as a result of vertical and transverse cracks combination. Eventually, the vertical crack extended to the upper-end face of specimen C1, and the transverse crack almost cut across the entire specimen C1 (see Fig 5c). After temperature shocks, the crack volume was 24.169 mm$^3$, 34.81 times increase, and the maximal crack width became to 60 μm, an increase of 50%.

In Fig 5d, it can be seen that several initial microcracks are scattered in coal specimen D1. After heating-cooling shock, the crack volume of specimen D1 increased from 3.738 mm$^3$ to 49.439 mm$^3$, a 12.23 times increase, and the maximal crack width raised from 50 μm to 70 μm, a 40% increment. Under the thermal stress effect from heating-cooling shock, initial cracks expanded and interconnected to eventually form a circular crack network paralleling to parallel beddings in specimen D1. The formed three crack networks were approximately perpendicular to the parallel beddings.

Admittedly, coal is a type of heterogeneous material made up of organic matter, pores, cement, microcracks, and fissures. When coal is subjected to thermal shocks, non-uniform deformations and thermal stresses are produced due to the difference in thermal expansion coefficients, leading to local stress concentrations at the tips of cracks [57]. Once thermal stresses induced by thermal shocks exceed the mechanical strength of coal, initial cracks will extend forward and generate some new microcracks in weak areas [58, 59]. Therefore, under the effect of thermal shocks, the extension and growth of microcracks significantly enhanced their connectivity and eventually improved the coal permeability (see Fig 4).

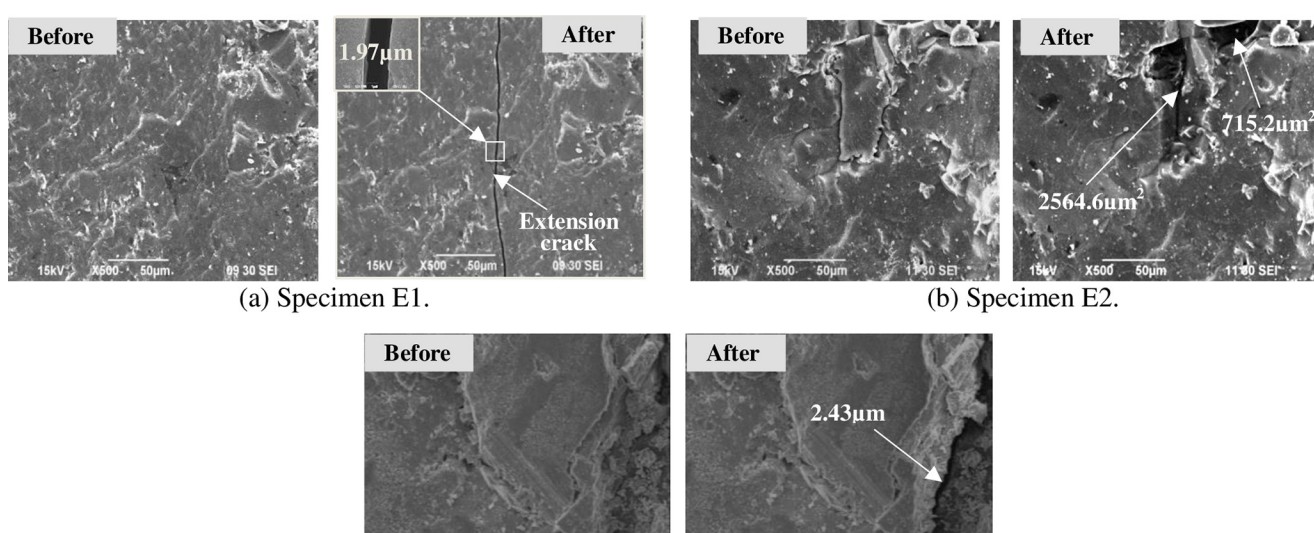

(a) Specimen E1.  (b) Specimen E2.

(c) Specimen E3.

**Fig 6. SEM images of anthracite coal particles before and after cooling shock.**

## 3.3 SEM Results under thermal shocks

To clearly observe the propagation and growth of microcracks, SEM tests were conducted in laboratory under two kinds of thermal shocks (i.e. cooling shock and heating-cooling shock). Four groups of coal specimens were subjected to these thermal shocks, two groups (groups E and F) to cooling shocks, and two groups (groups G and H) to heating-cooling shocks. For a high testing resolution, all coal specimens for thermal shocks were coal particles of 3 mm to 6 mm size. In Figs 6 and 7, "Before" means "Before the cooling shock", and "After" means "After the cooling shock". In Figs 8 and 9, "Before" means "Before the heating-cooling shock", and "After" means "After the heating-cooling shock".

**3.3.1. SEM images under cooling shocks.** Coal specimens in groups E and F were subjected to cooling shocks. The coal specimens in group E numbered as E1, E2 and E3 were anthracite coal, while the coal specimens in group F numbered as F1, F2 and F3 were bituminous coal.

Fig 6 illustrates SEM images of the anthracite specimens before and after cooling shocks. After the cooling shock, a single microcrack with a width of about 1.97 μm was formed in specimen E1 (Fig 6a), indicating that fracture easily occurs in the weak area of low cohesion. Two small coal grains with 2564.6 μm$^2$ and 715.2 μm$^2$ area were peeled off from specimen E2 due to coal matrix dislocation and shrinkage under the cooling shock. Specimen E3 was relatively rough and loose compared to the other two specimens. Similar to specimen E1, a new microcrack of approximately 2.43 μm width was also generated along the weak surfaces of specimen E3.

Fig 7 shows SEM images of the bituminous coal particles before and after cooling shock. It is clear that the development degree of endogenous cracks is closely associated to the metamorphic rank of coal. Generally, the development of internal cracks in bituminous coal is better than that in anthracite [60]. Thus, the number of pre-existing cracks of coal specimens in group F is larger than that of specimens in group E (see Figs 6 and 7). According to the SEM

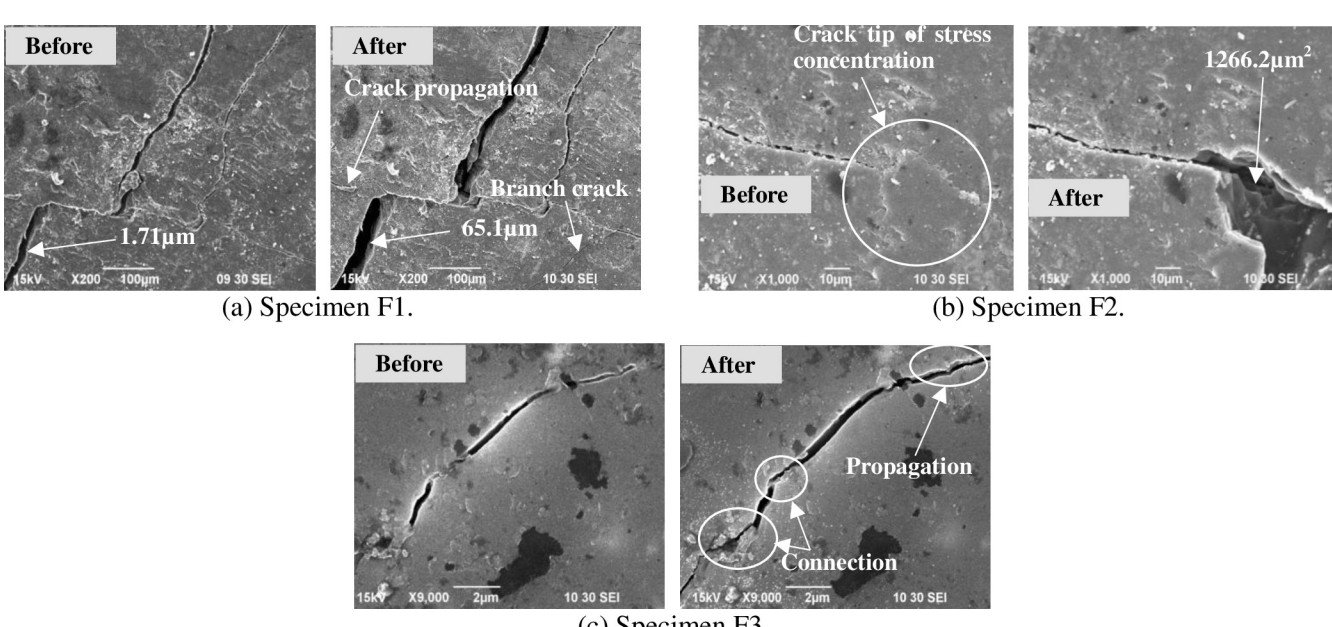

**Fig 7. SEM images of bituminous coal particles before and after cooling shock.**

images of specimen F1 (Fig 7a), the crack width increased from 17.1 μm to 65.1 μm, 3.81 times of the crack width before the cooling shock. In addition, a new microcrack branch was formed along the pre-existing crack boundary. Furthermore, stress concentration at the tips of a crack can also lead to the detachment of coal grains. A coal grain with an area of 1266.2 μm$^2$ peeled off from specimen F2 because the maximum stress caused by the cooling shock exceeded the coal mechanical strength. Under thermal stress, the microcrack in Fig 7b expanded forward and eventually formed a macroscopic crack until the crack tip was blunt. According to specimen F3, a few discontinuous microcracks co-existed in the same region. These microcracks extended forward and interconnected to each other, forming a new macrocrack (see Fig 7c).

**3.3.2. SEM images under heating-cooling shocks.** Coal specimens in groups G and H were subjected to heating-cooling shocks. The coal specimens in group G numbered as G1, G2 and G3 were anthracite coal, while the coal specimens in group H numbered as H1, H2 and H3 were bituminous coal.

Fig 8 shows SEM images of the anthracite specimens before and after heating-cooling shock. After heating-cooling shock, complex dendritic microcracks formed on the surfaces of the coal specimens due to the high temperature difference. Compared to the simple microcracks caused by cooling shock, the secondary microcracks caused by the heating-cooling shock were more bifurcate and zigzag. As shown in Fig 8a and 8b, there were no microcracks on the surfaces of specimens G1 and G2 before the heating-cooling shock, but several microcrack branches were produced after the heating-cooling shock. In Fig 8c, there were two new generated microcracks, symmetrically located on both sides of the main fracture in specimen G3. This kind of cracks are known as called wing cracks [61].

Fig 9 displays SEM images of the bituminous coal particles before and after heating-cooling shock. Similar to the SEM images in Fig 7, natural microcracks in bituminous coal were more developed than that in anthracite and filled with fillers and cement formations. In cording to

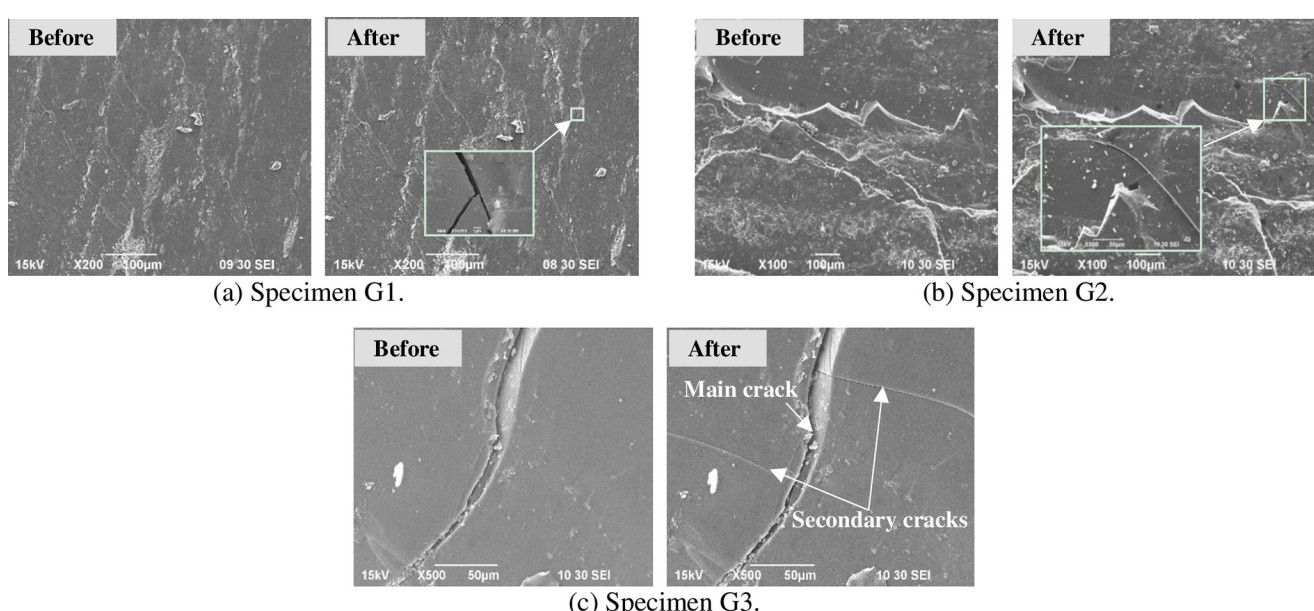

**Fig 8. SEM images of anthracite coal particles before and after the heating-cooling shock.**

the SEM images of specimen H1, after the heating-cooling shock, a few discontinuous micro-cracks interconnected each other and finally formed a new macrocrack of 2.75 μm width (see Fig 9a). This new macrocrack jaggedly extended forward with an intrusion of extra micro-crack. From the SEM images of specimen H2, an initial microcrack of 0.89 μm width grew into a 2.65 μm width main crack through the fracture of cement formations and the shedding

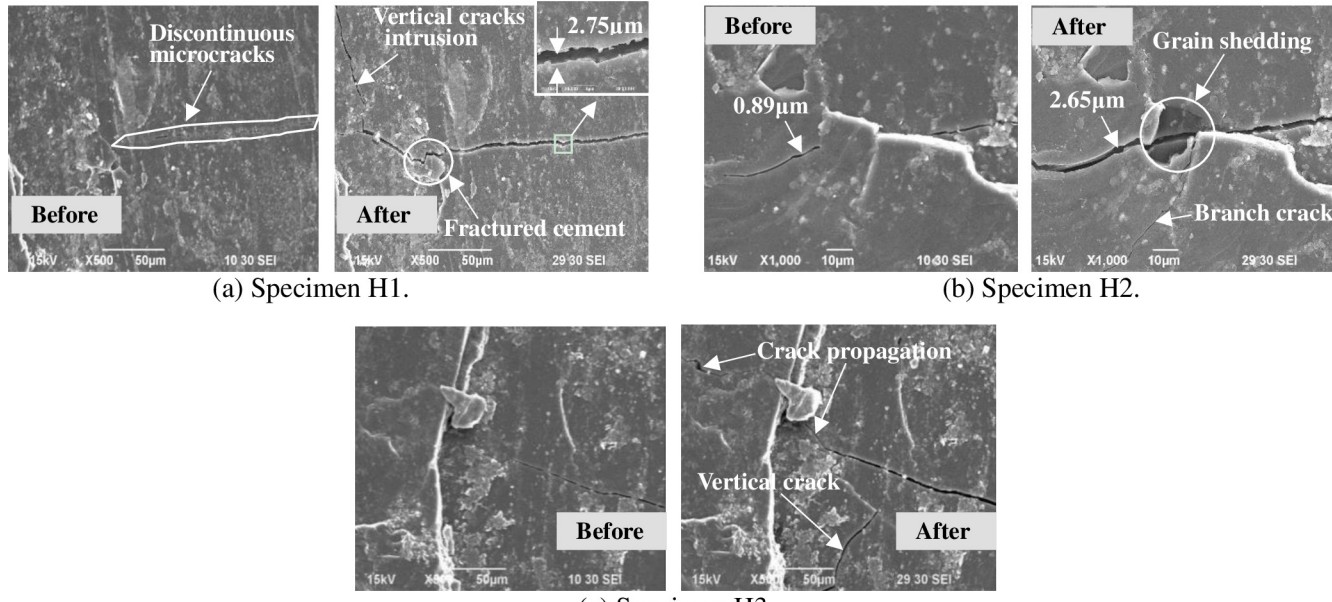

**Fig 9. SEM images of bituminous coal particles before and after heating-cooling shock.**

of the coal grains (Fig 9b). A bifurcation microcrack also formed at the middle of the main crack. From the SEM images of specimen H3, a single microcrack spread in a zigzag manner in the coal grains and cement formations area, and finally formed a complex main crack.

From the above result analysis, under thermal shocks effect, the propagation rule of microcracks obtained by analyzing CT images was consistent with that by analyzing SEM images. Previously, studies to investigate the internal structure of microcracks in geological materials were carried out by using CT and SEM images [43, 60, 62]. For instance, Roslin [43] used CT scan and SEM techniques to study the characteristic of cleat structures in coal in order to predicate coal permeability with Lattice Boltzmann method. In this paper, CT scanning focused on the internal microcracks change and whereas SEM scanning focused on the surface microcracks change. Although the size of the coal specimens used for CT scanning was different from that for SEM scanning, these two testing means essentially reflected a same variation trend of microcracks due to thermal shocks. Therefore, the results from CT and SEM scans are realistic and reliable in describing microcrack propagation under thermal shocks.

## 4. Crack propagation criterion of coal

Studies have that once thermal stress caused by thermal shocks in coal specimens exceeds coal peak strength, crack propagation and coal failure occur [63, 64]. To explain the propagation and growth of microcracks in Figs 6, 7, 8 and 9, it is imperative to determine the stress field near the crack tips and the failure conditions of microcracks. Cracks can be divided into three types based on stress states: opening mode (Type I), sliding mode (Type II), and tearing mode (Type III) [65]. When exposed to thermal shocks, the center oblique cracks in the coal specimens were generally subjected to tensile or compressive stresses, which matches the I-II complex crack stress model.

According to thermodynamic theory, thermal stresses can be produced in coal specimens during thermal shocks; compressive thermal stress can be caused by cooling shock treatment, while tensile thermal stress can be caused by heating shock treatment. On the basis of crack propagation criterion of coal, a theoretical model is proposed in this paper. In this model, an oblique crack with a length of 2$a$ is subjected to a load of biaxial stretching, as shown in Fig 10.

By transforming the stress coordinate, the expressions of normal stress $\sigma_y$ and shear stress $\tau_{xy}$ in the principal crack plane can be written as follows:

$$\sigma_y = \sigma_1 \sin^2\beta + \sigma_3 \cos^2\beta \qquad (2)$$

$$\tau_{xy} = (\sigma_1 - \sigma_3)\sin\beta\cos\beta \qquad (3)$$

where $\sigma_1$ and $\sigma_2$ are the biaxial thermal stresses, $\beta$ denotes the angle between the crack length direction and the load direction; and $\sigma_y$ and $\tau_{xy}$ stands for normal stress and shear stress in principal crack plane, respectively.

The thermal stresses generated in the coal specimen can be roughly calculated by:

$$\sigma_i = \alpha_i E_i \Delta T \delta_i \qquad (4)$$

where $\sigma_i$ is the thermal stress, $\alpha_i$ denotes the linear expansion coefficient of coal, $E_i$ is the elastic modulus of coal, and $\delta_i$ stands for Kronecker symbol.

Due to the anisotropy of coal, the thermal expansion coefficient $\alpha_i$ and the elastic modulus $E_i$ are different in vertical and parallel bedding directions, and therefore $\sigma_1 \neq \sigma_3$. Supposing $\sigma_1 > \sigma_3$ and $\sigma_1 = \sigma$, then the thermal stress $\sigma_3 = k\sigma_1 = k\sigma$ $(0<k<1)$ in the transverse direction can be obtained. Using Eqs (2) and (3), the normal stress ($\sigma_y$) and shear stress ($\tau_{xy}$) acting on the

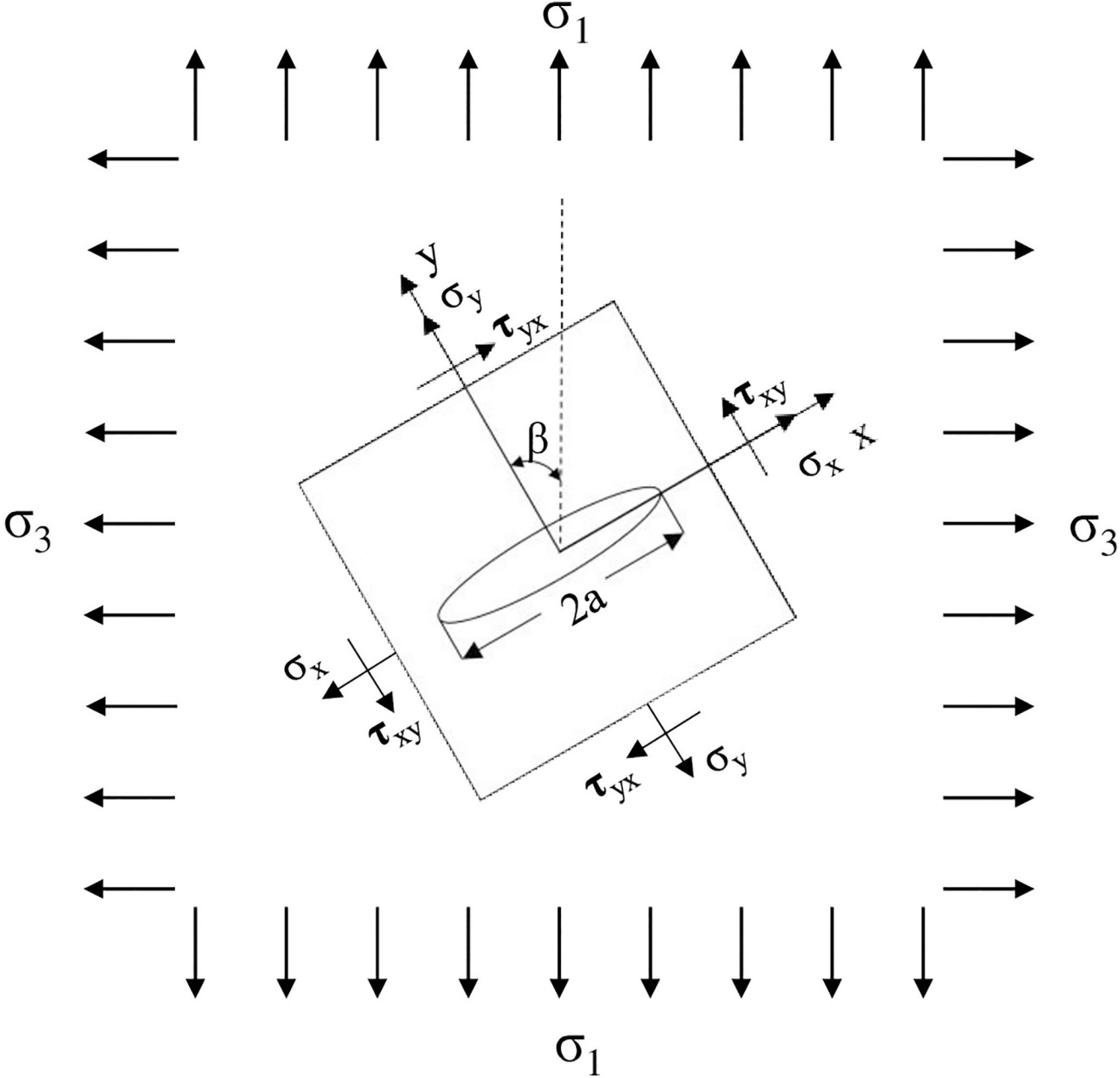

**Fig 10. Stress model of central oblique crack [66].**

crack can be expressed as follows:

$$\sigma_y = \sigma(\sin^2 \beta + k \cos^2 \beta) \tag{5}$$

$$\tau_{xy} = (1 - k)\sigma \sin \beta \cos \beta \tag{6}$$

The crack expands outward because of the tension effect caused by a heating shock. This keeps the crack open even after heating shock. In this case, the friction between the two walls of the crack can be ignored, and the shear stress acting on the crack is represented by $\tau_{xy}$ in

(Eq 6). The open crack closes because it shrinks inward when subjected to cooling shock. To calculate shear stress of the crack, the friction between the two walls of the crack should be taken into account. Under the action of cooling shock, the shear stress acting on the crack can be expressed as:

$$\tau'_{xy} = (1 - k)\sigma \sin \beta \cos \beta - \mu\sigma(\sin^2 \beta + k \cos^2 \beta) \tag{7}$$

where $\tau'_{xy}$ denotes the effective shear stress, and $\mu$ stands for the friction factor [67].

According to Eqs (6) and (7), the shear stress caused by heating shock exceeds that by cooling shock, indicating that heating shock is more favorable to the propagation of microcracks in coal. Based on the knowledge of fracture mechanics [68], the stress model of the center oblique crack shown in Fig 9 can be considered a combination of two stress models Type I and II. The stress intensity factor (SIF) of Type I crack is expressed as follows:

$$K_I = Y_1 \sigma \sqrt{\pi a}(\sin^2 \beta + k \cos^2 \beta) \tag{8}$$

Under tensile and compressive conditions, the SIFs of Type II crack can be expressed as follows:

$$K_{II} = (1 - k)\sigma\sqrt{\pi a} \sin \beta \cos \beta \tag{9}$$

$$K_{II} = Y_2\sqrt{\pi a}[(1 - k)\sigma \sin \beta \cos \beta - \mu\sigma(\sin^2 \beta + k \cos^2 \beta)] \tag{10}$$

where $Y_1$ and $Y_2$ represent the shape factors of Type I and Type II cracks, respectively, and correspond to different crack shapes; $a$ denotes the half length of the crack.

Under thermal shocks, a stress field formula of the mixed model Type I-II can be reformed by the superposition of the stress fields of Type I and Type II. In a polar coordinate (Fig 11), the stress field at the crack tip area is described by the following equation:

$$\begin{pmatrix} \sigma_r \\ \sigma_\theta \\ \tau_\theta \end{pmatrix} = \frac{\cos\dfrac{\theta}{2}}{2\sqrt{2\pi r}} \begin{pmatrix} K_I(3 - \cos\theta) + \tan\dfrac{\theta}{2}K_{II}(3\cos\theta - 1) \\ K_I(1 + \cos\theta) - 3K_{II}\sin\theta \\ K_I\sin\theta + K_{II}(3\cos\theta - 1) \end{pmatrix} \tag{11}$$

In accordance with the maximum circumferential stress criterion [68], the cracking angle is obtained using the following equation:

$$\frac{\partial \sigma_\theta}{\partial \theta} = 0 \tag{12}$$

Substituting the expression of $\sigma_\theta$ in (Eq 11) into (Eq 12), the expressions of the cracking angle ($\theta_0$) and the crack propagation criterion are determined by:

$$\theta_0 = \arccos\frac{3K_{II}^{\,2} + \sqrt{K_I^{\,4} + 8K_I^{\,2}K_{II}^{\,2}}}{K_I^{\,2} + 9K_{II}^{\,2}} \tag{13}$$

$$\cos\frac{\theta_0}{2}\left(K_I \cos^2\frac{\theta_0}{2} - \frac{3}{2}K_{II}\sin\theta_0\right) = K_{IC} \tag{14}$$

where $K_{IC}$ represents fracture toughness and it can be measured experimentally.

Under the action of thermal stresses, according to Eqs (13) and (14), the cracking direction and cracking criterion of coal can be derived using the values of $K_I$ and $K_{II}$. During thermal

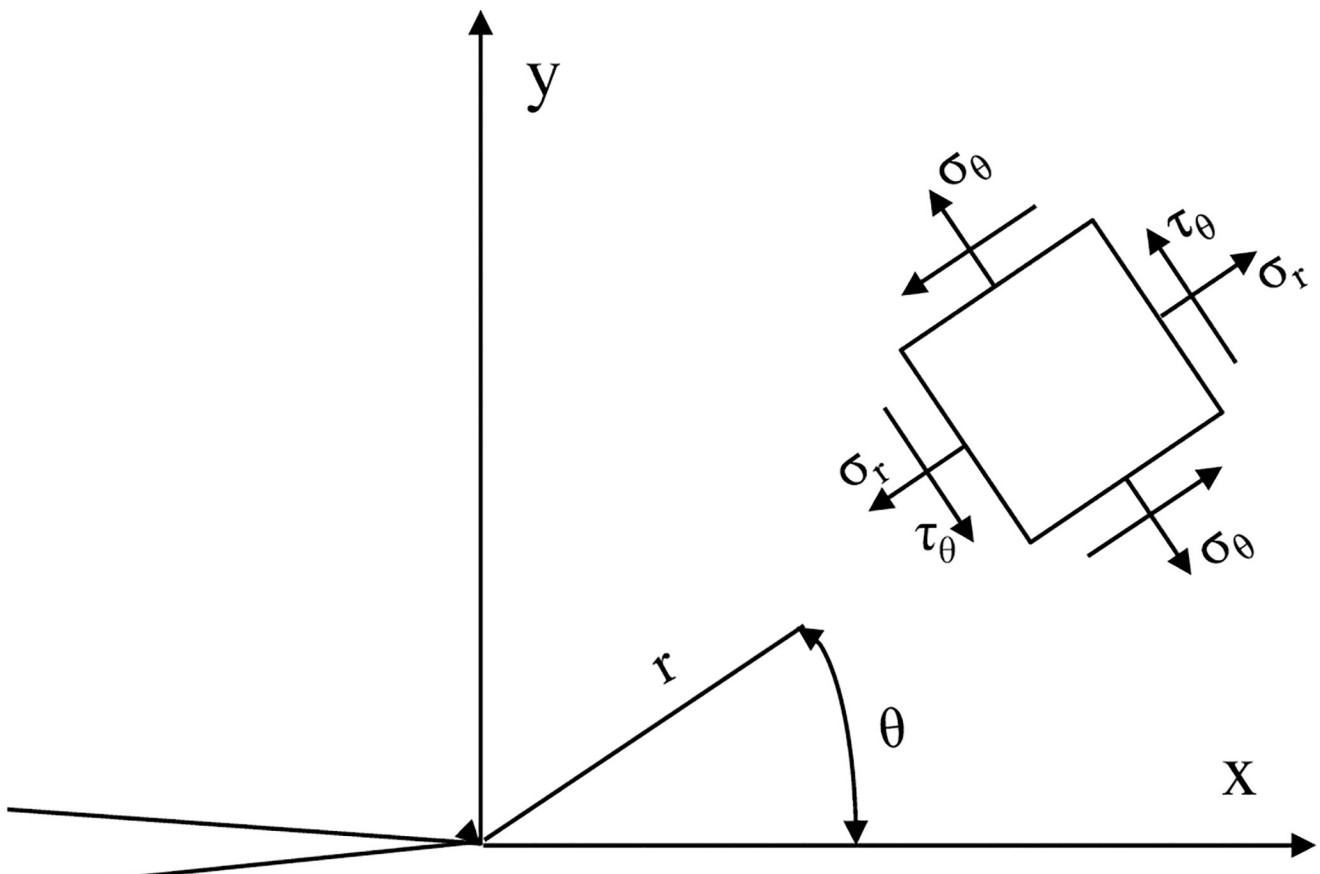

**Fig 11. Crack-tip stress field in a polar coordinate.**

shocks, thermal stresses induce heterogeneous expansion and contraction of coal matrix, resulting in local stress concentrations in the coal specimens. When the thermal stresses exceeded the coal peak strength, expansion and growth of microcracks occurred, which enhanced microcrack connectivity and increased the permeability of coal specimens. The thermal stress caused by heating-cooling shocks exceeded that by cooling shocks because of the higher temperature difference in the heating-cooling shocks. Thus, the propagation and connection of microcracks are more sufficient and more effective under heating-cooling shocks, leading in a larger increase in coal permeability.

## 5. Conclusions

The following conclusions are drawn in this study.

1. Exposure of coal specimens to thermal shocks increased the permeability of coal at various temperatures. The 3D CT scanning images showed that thermal shocks enhanced crack propagation in coal, especially when temperature difference increased. Thermal shocks increased the expansion and connection among cracks, widened the gas flow paths and effectively improved the permeability of coal.

2. SEM images showed that simple, bifurcated and wing cracks were produced on the surfaces of the coal specimens during thermal shocks. The number of new cracks caused by heating-cooling shock was larger than that caused by cooling shock, and the crack propagation path caused by heating-cooling shock was more complex than that caused by cooling shock. Thus, heating-cooling shock is superior to cooling shock in damaging and breaking coal.

3. Based on the propagation characteristics of cracks, a crack propagation criterion was proposed to theoretically explain the mechanism by which thermal shocks break coal and increase its permeability. Furthermore, the results in this work can enrich the boosting of CBM production in future.

## Supporting information

**S1 Data.**
(OPJ)

**S1 Table. Test results of permeability of coal samples before and after thermal shock.**
(XLSX)

## Author Contributions

**Conceptualization:** Dengke Wang.

**Data curation:** Hongtu Zhang, Chong Yu, Jianhua Fu.

**Investigation:** Hongtu Zhang, Dengke Wang, Chong Yu, Jianping Wei, Shumin Liu.

**Methodology:** Dengke Wang.

**Resources:** Hongtu Zhang.

**Supervision:** Dengke Wang, Jianping Wei.

**Validation:** Dengke Wang, Shumin Liu, Jianhua Fu.

**Writing – original draft:** Hongtu Zhang.

**Writing – review & editing:** Dengke Wang, Chong Yu, Jianping Wei, Shumin Liu, Jianhua Fu.

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
