## [Decision Letter · Decision Letter 0]

17 Feb 2020

PONE-D-20-03020

Microcrack evolution and permeability enhancement due to thermal shocks in coal

PLOS ONE

Dear Dr. Wang,

Thank you for submitting your manuscript to PLOS ONE. After careful consideration, we feel that it has merit but does not fully meet PLOS ONE’s publication criteria as it currently stands. Therefore, we invite you to submit a revised version of the manuscript that addresses the points raised during the review process.

English and presentations should be significantly improved before acceptance. More details on experimental process should be given.Resonability and consistency of these results in different scales should be discussed.

We would appreciate receiving your revised manuscript by Apr 02 2020 11:59PM. To enhance the reproducibility of your results, we recommend that if applicable you deposit your laboratory protocols in protocols.io, where a protocol can be assigned its own identifier (DOI) such that it can be cited independently in the future. For instructions see: http://journals.plos.org/plosone/s/submission-guidelines#loc-laboratory-protocols

We look forward to receiving your revised manuscript.

Kind regards,

Jianguo Wang, PhD

Academic Editor

PLOS ONE

Journal Requirements:

Reviewers' comments:

Reviewer's Responses to Questions

**Comments to the Author**

1. Is the manuscript technically sound, and do the data support the conclusions?

Reviewer #1: Yes

Reviewer #2: Yes

2. Has the statistical analysis been performed appropriately and rigorously? 

Reviewer #1: Yes

Reviewer #2: Yes

3. Have the authors made all data underlying the findings in their manuscript fully available?

Reviewer #1: Yes

Reviewer #2: Yes

4. Is the manuscript presented in an intelligible fashion and written in standard English?

Reviewer #1: Yes

Reviewer #2: Yes

5. Review Comments to the Author

Reviewer #1: I recommend it for publication in this Journal subjected to following comments and clarifications.

1. The format of the references needs to be modified to meet the publication requirements of PLoS ONE.

2. The technical parameters and measure accuracies of the testing apparatus should be listed in the text.

3. What’s the meaning of “The prepared coal samples were placed in a drying oven for the purpose of dry-seal preservation under temperature of 60 �C”? Didn’t the coal samples need to be dried before testing.

4. Equation (1) should be cited in the manuscript.

Reviewer #2: In this manuscript, the evolution of the crack extension and the enhancement of coal permeability were studied under the effect of thermal shocks. The conclusions yielded in this paper are important for CBM exploitation. I recommend accepting this manuscript before the following problems are addressed.

1) The authors should explain why the sizes of the coal samples tested in CT and SEM scanning tests are different.

2) On page 4, line 26, “120℃ -120 ℃” should be “120℃ - -120 ℃”, “150℃ -150 ℃” should be “150℃ - -150 ℃”, “180℃ -180 ℃” should be “180℃ - -180 ℃”.

3) The experimental functions, the maker, and technical parameters and scanning resolution of the CT scanning equipment should be provided in the context.

4) Please explain why the period of thermal shocks was set to 2 hours.

5) “temperature impact” should be “thermal shock”.

6. PLOS authors have the option to publish the peer review history of their article (what does this mean?). If published, this will include your full peer review and any attached files.

Reviewer #1: No

Reviewer #2: No

---

## [Author Response · Author response to Decision Letter 0]

26 Mar 2020

1. English and presentations should be significantly improved before acceptance.

Reply: We have already improved English expression and corrected mistakes in grammar. 

2. More details on experimental process should be given.

Reply: The detailed experimental process has already added in the paper. 

3. Resonability and consistency of these results in different scales should be discussed.

Reply: The discussion of the reasonability and consistency of the results from CT and SEM scans is added at the end of Section 3.3.2. 

Reviewer #1: I recommend it for publication in this Journal subjected to following comments and clarifications.

(1) The format of the references needs to be modified to meet the publication requirements of PLoS ONE.

Reply: The references have modified.

(2) The technical parameters and measure accuracies of the testing apparatus should be listed in the text.

Reply: The technical parameters and measure accuracies of the testing apparatus have added in our paper.

(3) What’s the meaning of “The prepared coal samples were placed in a drying oven for the purpose of dry-seal preservation under temperature of 60 ℃”? Didn’t the coal samples need to be dried before testing.

Reply: To avoid frost heaving effect from internal moisture during thermal impacts, all examined coal samples were thoroughly dried using an oven at 105 ℃ for 24 h. “The prepared coal samples were placed in a drying oven for the purpose of dry-seal preservation under temperature of 60 ℃” means that 60 ℃ is the preservation temperature of the coal samples, not the dry temperature. 

(4) Equation (1) should be cited in the manuscript.

Reply: Equation (1) has been cited. 

Reviewer #2: In this manuscript, the evolution of the crack extension and the enhancement of coal permeability were studied under the effect of thermal shocks. The conclusions yielded in this paper are important for CBM exploitation. I recommend accepting this manuscript before the following problems are addressed.

(1) The authors should explain why the sizes of the coal samples tested in CT and SEM scanning tests are different.

Reply: In this paper, CT scanning was focused on the change of internal microcracks and SEM scanning was focused on the change of surface microcracks. Although the size of the coal samples used for CT scanning was different from that for SEM scanning, these two testing means actually reflected a same change trend of microcracks caused by thermal shocks. Therefore, the results from CT and SEM scans were reasonable and consistent in describing the microcrack propagation under thermal shocks. 

(2) On page 4, line 26, “120℃ -120 ℃” should be “120℃ - -120 ℃”, “150℃ -150 ℃” should be “150℃ - -150 ℃”, “180℃ -180 ℃” should be “180℃ - -180 ℃”.

Reply: We have corrected this mistake in our manuscript. 

(3) The experimental functions, the maker, and technical parameters and scanning resolution of the CT scanning equipment should be provided in the context.

Reply: The experimental functions, the maker, and technical parameters and scanning resolution of the CT scanning equipment have added to our paper. 

(4) Please explain why the period of thermal shocks was set to 2 hours.

Reply: According to thermodynamic theory, the coal samples of ϕ50×50mm size were subjected to thermal shocks for 2 hours is keep all parts in the coal samples can be affected by thermal shock. 

(5) “temperature impact” should be “thermal shock”.

Reply: We have replaced “temperature impact” with “thermal shock” in our manuscript.

---

## [Decision Letter · Decision Letter 1]

9 Apr 2020

Microcrack evolution and permeability enhancement due to thermal shocks in coal

PONE-D-20-03020R1

Dear Dr. Wang,

We are pleased to inform you that your manuscript has been judged scientifically suitable for publication and will be formally accepted for publication once it complies with all outstanding technical requirements.

With kind regards,

Jianguo Wang, PhD

Academic Editor

PLOS ONE

Additional Editor Comments (optional):

This manuscript is accetable technically but English and presentations should be further improved to meet the requirements for publication.

Reviewers' comments:

Reviewer's Responses to Questions

**Comments to the Author**

1. If the authors have adequately addressed your comments raised in a previous round of review and you feel that this manuscript is now acceptable for publication, you may indicate that here to bypass the “Comments to the Author” section, enter your conflict of interest statement in the “Confidential to Editor” section, and submit your "Accept" recommendation.

Reviewer #1: All comments have been addressed

Reviewer #2: All comments have been addressed

2. Is the manuscript technically sound, and do the data support the conclusions?

Reviewer #1: Yes

Reviewer #2: Yes

3. Has the statistical analysis been performed appropriately and rigorously? 

Reviewer #1: Yes

Reviewer #2: Yes

4. Have the authors made all data underlying the findings in their manuscript fully available?

Reviewer #1: Yes

Reviewer #2: Yes

5. Is the manuscript presented in an intelligible fashion and written in standard English?

Reviewer #1: Yes

Reviewer #2: Yes

6. Review Comments to the Author

Reviewer #1: All my concerns have been addressed. Thus, I suggest this revised manuscript to be published in PLOS ONE.

Reviewer #2: (No Response)

7. PLOS authors have the option to publish the peer review history of their article (what does this mean?). If published, this will include your full peer review and any attached files.

Reviewer #1: No

Reviewer #2: No

---

## [Editor Report · Acceptance letter]

8 May 2020

PONE-D-20-03020R1 

Microcrack evolution and permeability enhancement due to thermal shocks in coal 

Dear Dr. Wang:

I am pleased to inform you that your manuscript has been deemed suitable for publication in PLOS ONE. Congratulations! Your manuscript is now with our production department. 

With kind regards,

on behalf of

Dr. Jianguo Wang 

Academic Editor

PLOS ONE